# The determinant factors for the adoption of CRM in the Palestinian SMEs: The moderating effect of firm size

**Omar Hasan Salah** [1]*, **Zawiyah Mohammad Yusof**[2], **Hazura Mohamed**[2]

**1** The College of Business and Economics, Palestine Technical University, Kadoorie, Tulkarm, Palestine,
**2** Department at the Universiti Kebangsaan, School of Information Science and Technology, Bangi, Malaysia

* Omar_salah79@hotmail.com

## Abstract

CRM adoption can provide innumerable benefits to the SMEs performance, including solving customer problems in a timely manner, enhancing customer satisfaction by appointing an expert to solve issues and queries, and the like. This study aims to examine the moderating effects of the firm size in the adoption of CRM in the Palestinian SMEs. A quantitative approach was used to investigate the relationships between the variables, which are compatibility, IT infrastructure, complexity, relative advantage, security, top management support, customer pressure, and competitive pressure. A questionnaire was designed to collect data from 420 SMEs in Palestine. A total of 331 respondents completed and returned the survey. The Partial Least Square-Structural Equation Model (PLS-SEM) approach was used to assess both the measurement and structural models. The Diffusion of Innovation Theory (DOI) and Technology, Organization, and Environment Framework (TOE) framework were employed to identify the determinant factors from the technological, organizational, and environmental perspectives. The findings and conclusions of this study provide show that the moderating effect of firm size has significant effect compatibility, top management support, customer pressure, and IT infrastructure factors.

**Data Availability Statement:** All relevant data are within the manuscript and Supporting Information files.

**Funding:** The study is financially supported by Research Grant GUP-2017-046, Universiti

## 1. Introduction

Information and communication technology (ICT) provides small and medium enterprises (SMEs) with strategic advantages in achieving goals and improved competitive edge. The term ICT is a combination of two terms: information technology (IT) and communications. Information technology refers to any computing technology, such as networks, hardware, software, communication devices, communication infrastructure [1], systems and solutions, and the internet IT can lower production and labor costs, add value to products and services, and increase a company's competitive advantage. Some studies and reports have shown that IT is a means that could enhance the business process [2]. A system, or sometimes referred to as technology such as customer relationship management (CRM) systems, would be able to help in the materialization of such a determination [1,2]. Both scholars and practitioners have given CRM attention since the past two decades [3,4] due to the wide range of benefits it offers and

Kebangsaan Malay- sia, and the Public Services Department of Malaysia.

**Competing interests:** The authors have declared that no competing interests exist.

many untold advantages it has. Such pertinent characteristics of CRM has made the system considered as an essential part of current businesses that seek to increase revenues and maintaining the company's performance [5,6].

Customer relationship management (CRM) refers to the practices, strategies, and technologies used by firms and businesses for the management and analysis of customer interactions and data through the customer lifecycle [7,8]. It is the concept of using process, information, technology, and people to manage the organization's interaction with customers [9]. The technology could be seen as an approach to marketing that has its origins in relationship marketing.

There are various definitions accorded to CRM. To some scholars, CRM is an organizational approach for understanding the behaviors of the customers in order to attract and retain them [10]. In this regard, CRM is essential for managing the relationship of the organization with customers that relate to the overall process of marketing [11]. As the fourth most used tool in business, which holds the key to understanding to create a sustainable customer relationship, CRM is the center of obtaining and maximizing the number of loyal customers [12]. Thus, CRM has gained the reputation of being a significant business tool from both academics and practitioners, although there are challenges in the adoption [13]. In relation to this, CRM is considered as technology when it serves as a tool. This is revealed from the technical definition where CRM emphasis is on the information system [14]. The technical definition of CRM by [15] indicates that CRM technology is a method that significantly uses information technology (IT), particularly the databases and the Internet, in controlling and integrating the entire company's marketing effort and automating the specific customer-organization relationships [16]. This would, in turn, help the company to obtain and retain valuable information vis-à-vis increased knowledge. A company with sufficient and appropriate knowledge is more successful and equipped compared to the one with insufficient information to strategize their business. As technology, the CRM system consists of three major parts. These are: i) technologies employed for external customers operations which bring about a two-way communication between the customers and the firm, ii) technologies employed for internal operations like marketing, sales, and customer service, which direct activities to automation and facilitating activities and, iii) technologies that drive other technologies and allow firms to analysis and dissemination of data across organizational departments [17].

On the other hand, when the term is defined broadly (i.e., strategic definition), CRM is referred to as a way of managing and facilitating business processes and activities of the company [14]. The strategic definition usually inclusive of the supply chain, sales, outstanding orders, service and repair, unresolved issues, customer-organization interactions, customer service inclusive of stakeholders, and labor requirements.

## 2. Customer Relationship Management (CRM)

Customer Relationship Management (CRM) is explained as a concept of use process, information, technology, and people to manage the organizations' interactions with its customers [9]. By applying efficient and effective CRM technology, CRM employees in SMEs could retain existing customers, attract new customers, foster customer satisfaction, and increase profitability. The system is an enabler in achieving the objectives of gathering, categorizing, and saving the required customer data. Integration technology enables the development of customer-organization relationships through broader information of customer behavior.

As technology, the CRM system consists of three major parts. These are: i) technologies employed for external customers operations which bring about a two-way communication between the customers and the firm, ii) technologies employed for internal operations like

marketing, sales, and customer service, which direct activities to automation and facilitating activities and, iii) technologies that drive other technologies and allow firms to analysis and dissemination of data across organizational departments [17].

## 2.1. CRM adoption

The increasing requirement for technology over the past thirty years and the increasing failures of system adoption has shifted focus to predict the use of a system [18]. Adoption is a stage that an individual goes through to decide to accept/reject a new idea/innovation [19]. The term describes the acceptance to use something novel or different.

CRM adoption is the willingness of the companies to adopt CRM in order to achieve different objectives such as enhancing a relationship with customers, understanding customer requirements, increasing customer loyalty, and improving revenues [20]. [18] revealed that user acceptance/rejection of technology is one of the main challenges that are faced by organizations in IS adoption. Technology rejection arises because of the lack of knowledge of new technology among adopters, lack of prediction of the consequences that the innovation will bring, or the status-conferring aspect of the technology [19]. The adoption of a CRM system is used to provide employees with the processing of customers' information [21].

## 2.2. CRM solution providers

CRM is characterized as costly and complex innovation, and as such, it needs integrated information systems, costly infrastructure facilitation, and advanced technological skills and knowledge for its implementation and usage (Laketa et al. 2015) [22]. CRM technology should be authenticated before use and based on customer's preferences, which are important in implementing a successful CRM. "This software consists of many multi-functional solutions that can sort out the information management of clients as well as the automation interaction processes of clients" (Faed et al. 2010) [23].

CRM systems differ from one to the next as CRM vendors follow varying technological standards for their development [24]. CRM solutions investments in the form of software, hardware, and services provide support to office functions in terms of marketing, selling, and services [25,26]. In relation to this, Rosman and Stuhura (2013) [27] noted that CRM technology relates to different organizational departments together, allowing the firms to interact efficiently with their customers. Among the main advantages offered by CRM technology is its capability of integrating major business functions, and the integration of customer service functions may exemplify this into one information system [28]. Hence, there are three CRM Solution Providers: SAP, Oracle Corporation, and Microsoft Corporation.

## 2.3. The drawback of CRM

Although the benefit of CRM adoption is acknowledged in helping companies or enterprises to success [29] but the effect of CRM technology adoption is more on the sales forces managing customer relationships, rather than selling products [30]. The initiative enables firms to keep track of issues that the customers face, oversee service response, and appropriate customer inquiries to an expert to answer [31]. Thus, the CRM system creation should be customer-centric from the beginning of system design.

Despite CRM has gained significant attention, it still suffers from conceptual and methodological flaws, as asserted by [32]. Inadequate studies have been carried out. Furthermore, these studies only focus on gaining a competitive advantage. Factors determining the success or failure of its adoption with respect to the developing countries [20,21] are yet to be undertaken. The root cause for the high failure rate is yet to be revealed [4,23].

SMEs in developing countries need to be competent since they are the backbone of the economy [24,25]. They are the core of entrepreneurial activity and innovative entities that facilitate new business operations and play a vital role by providing employment opportunities and boost economic development. Although CRM adoption is seen as capable of reaping such an objective, many SMEs are experiencing problems since there is no appropriate framework to guide the initiative [32].

## 2.4. Requirement for the adoption of CRM

The intensity of globalization, coupled with the increasing competition and ICT development, companies in the developing countries have been forced to concentrate on CRM for maximization of revenues. In light of this, SMEs have a key, and as such, they should be supported as agents of structural change, reducing marginalization and achieving the equitable distribution of income [33].

The Organization for Economic Co-operation and Development (OECD 2005) reports that SMEs constitute 95% of the total number of businesses. The majority of SMEs have been leveraging IT technology to support their processes. However, the IT adoption varies from that of major organizations owing to the fact that the former enterprises lack resources that the latter possesses [34].

An important characteristic and attribute that can be observed and understood from the previous works on CRM are that they have focused upon different industries such as the banking [22,35], the healthcare [36], and focused on the different levels and departments of organizations such as the call center [34,37]. There are limited studies that investigated the challenges of adopting a CRM strategy in SMEs in developing countries [33].

Currently, increasing ICT usage and development has opened up the Middle East countries economy into a global competition [32]. The CRM spending in this region has reached a rate of 10.7% [7]. It is "a customer-focused business strategy that dynamically integrates sales, marketing, and customer care service in order to create and add value for the company and its customers" [38,39]. The significance of CRM is reflected in many circumstances like increasing customer retention, making assumptions concerning the future transactions of customers, and providing an extensive overview of the customers and their requirements [40].

Moreover, customer relationship management (CRM) in the Middle East has become a key strategy for a small company, indicate the need for more SMEs to implement for effective business operations. It is evident that CRM also develops a high-performance strategy and facilitate value-added, technical, and innovative mechanisms to achieve the ultimate aim of obtaining a competitive edge over competitors [1,41].

CRM development took its root in the West but was soon followed by other countries in the other part of the world, especially the developing ones. However, not many studies were undertaken in the Middle East [32] except for a few studies on CRM in emerging markets [42]. This has resulted in the adoption of CRM in the Middle East is still lagging behind [32] except for Jordan, [35] where the market organization relationship has become the top predictors of organization performance. On the other hand, [43] links the CRM non-meeting of requirements in the Saudi Arabian service sector to the issues regarding customers' expectations and needs. Organizations need to plan out a roadmap for the adoption of a CRM system in order to achieve their goals.

## 3. The Palestine market and SMEs

There is no universal definition that can be accepted by all about the term. Different researchers carry different ideas regarding the capital layout, several employees, sales turnover, and

fixed capital investment upon which the definitions and categorizations of the concept are based on [44]. SMEs are categorized into classes based on some measurable quantitative indicators [45]. These enterprises form the heart of entrepreneurial activity and innovation as they play a key role in the economies of emerging nations through the provision of employment opportunities and increasing the development of the economy.

In the context of Palestine, the economy runs in an environment with internal and external risks challenges. The Palestinian territories have unemployment rates as high as 27% as of 2017 [46]. For example, in Gaza, the unemployment rate has reached 44%, in comparison to the West Bank (18%). Also, in 2017, only 41% of 15–29 years old employees remained active in the labor market, indicating the employees' high pessimistic attitude towards employment opportunities [47].

Nonetheless, the Palestinian Central Bureau of Statistics (PCBS) states that the ICT application adoption among Palestinians has experienced a gradual increase and gaining increasing acknowledgment [48]. ICT can function as a small business engine and benefit customer relationship management (CRM), which, in turn, could bring goals achievement to enhance competitiveness [3]. Since the majority of the Palestinian firms consist of SMEs [40,43], thus the adoption of CRM could materialize the goal achievement and enhance their competitiveness.

Palestine has a total of 14,359 enterprises where 99% of which belongs to the Palestinian SMEs. These SMEs employ 82% of the total workforce in the territory [49,50]. Tax revenue collected from SMEs forms 99% of business taxpayers aids to fund the nation's growth and development. Despite the small market Palestine has, SMEs remain as the backbone of economic growth and offer the most job opportunities [51] to the people with hope that could help in eradicating poverty [52]. Only a small number of these SMEs have direct access to foreign markets as the result of Israel's occupation, which has posed many problems and obstacles [43].

It is pertinent for SMEs in Palestine to adopt CRM for the efficient and effective management of customers, provides employees with easier processing of customers' information, retains the existing clients, attracts new and prospective clients, and promote and sustain customer satisfaction. In addition, CRM has been utilized as a tool for interaction management with customers, users, and to meet sales and financial requirements in day-to-day activities [9]. Thus, prior to the adoption of the initiative, factors which contribute to the success of adopting CRM must be made known. The examination of the shortcomings in CRM adoption in Palestinian SMEs could reveal the current status of CRM system usage and to overcome the shortcomings that arise. This would finally help to empower the economic situation of Palestine.

A review of literature has indicated that there is a lack of studies dedicated to the business strategies adopted by SMEs [50], be it in general on specifically in Palestine. One way to leverage the capability of SMEs is to embark on the CRM initiative. Prior to the adoption of the initiative, factors which contribute to the success of adopting CRM must be made known. Some of these factors are specific for a certain country only while some are common in nature. The examination of the shortcomings in CRM adoption in Palestinian SMEs could reveal the current status of CRM system usage and to overcome the shortcomings that arise. This would finally help to empower the economic situation of Palestine.

## 4. Justification for the moderating effects of firm size on those eight factors

The literature review shows (Table 1) that eight factors influence the adoption of CRM in SMEs. Three factors were adapted from the DOI model, and five factors were extracted from the literature, and TOE was used for factor classification. Table 1 provides the definitions and sources of the factors.

**Table 1. Factors influencing CRM adoption.**

| No | Factor | | Source (s) | Model/Theory |
|----|--------|---|------------|--------------|
| 1 | Compatibility | The level to which an innovation is viewed to match the current values, prior experiences, and the current requirements of potential users. | [53] | DOI |
| 2 | IT Infrastructure | infrastructure in IT is the whole collection of hardware, software, networks, data centers, facilities, and relevant equipment used for the development, testing, monitoring, managing, and supporting IS in an enterprise | [54] | Literature review |
| 3 | Complexity | The level to which an innovation is viewed as difficult to understand and utilize | [18] | DOI |
| 4 | Relative Advantage | The level to which an innovation is viewed as superior to the idea that came before it. | [19] | DOI |
| 5 | Security | The ability to protect consumers' information and transaction data to ensure their privacy | [55] | Literature review |
| 6 | Top Management Support | the level of support and understanding of top management concerning the functioning of IS and their contribution to its activities | [53] | Literature review |
| 7 | Customer Pressure | the end consumers' (primary stakeholder group) requests and requirements for the firm to enhance its environmental and social performance | [56] | Literature review |
| 8 | Competitive Pressure | The level of competitiveness in the industry within which the organization operates | [57] | Literature review |

These factors were drawn from an extensive range of frameworks in the literature. CRM practitioners then evaluate these factors in SMEs for verification and recommendation of new factors. These factors are expected to maximize the rate of CRM adoption among SMEs in developing countries, particularly Palestine.

## 5. Justifications for using PLS

Most researchers argue in favour of choosing PLS as the statistical means for testing structural equation models [58] because of it:

i. Fewer requests are made for sample size than other methods;

ii. Does not require regular -distributed input data;

iii. It can be applied for different sample sizes under the '10-times rule' method (Hair et al. 2014) [59]. (The 10-times rule is based on the rule that the sample size should be larger than ten times the maximum number of inner or outer pointing at an LV in the model [60];

iv. Can be used in the complex structural equation when large numbers of variables are involved;

v. Can handle both reflective and formative variables;

vi. is more suitable for theoretical development than theoretical testing; and

vii. It is particularly useful for prediction.

However, for prediction and explanation, when the phenomenon under study is relatively new, or when the theoretical model is complex with a large number of variables and indicator variables, a PLS approach is more preferred [59,61]. PLS is given the preference for several other SEM tools because PLS does not require a large sample size [58]. Also, PLS is more suitable when the objective of the study is causal predictive testing, rather than the testing of an entire theory

## 6. Moderating effect of firm size on CRM adoption

'Firm size' has been examined by several researchers in the field of innovation and has been considered to be a top indicator of organizational complexity [62]. The construct is measured through the number of employees in SMEs [6,22,27–41,43–54,63–65]. Despite the fact that a negative relationship has been revealed by some researchers between firm size and technology adoption (e.g., cloud computing) [56], a positive relationship has been supported by majority of studies in different contexts, such as e-commerce [66], mobile reservation systems [67], e-marketing [68], ICT innovations [69], as well as adoption of ICTs [70]. A few other studies reported the lack of a significant relationship between firm size and technology adoption (e.g., cloud computing) [56]. The size of the firm is the most critical adoption driver and lends strong support [71].

In smaller organizations, innovation is expected to be promoted by the availability of cross-functional cooperation [62], where such firms can adopt innovative practices easily through their flexibility in adopting changes in the environment rife with emerging market consumers [60,72]. Nevertheless, large firms have a higher likelihood to adopt new technology like CRM and e-commerce [27] and [61,62].

Literature dedicated to the IS field indicates that the successful adoption of technologies largely depends on CSFs (e.g., firm size) [71]. Some other researchers revealed that such CSFs do not significantly relate to the successful adoption of technology (Table 2). Meanwhile, [73] contended that firm size has a moderating effect on the level of knowledge acquisition.

## 7. Theoretical background

### 7.1 Diffusion of Innovation Theory

[87] developed the Diffusion of Innovation Theory (DOI) in the middle of the 20th century, with innovation seen as ideas, customs, or objects perceived by the individual or adopting units as something new. It was contended that several innovative product characteristics affect their adoption. These include their 'relative advantage', compatibility, complexity, divisibility, and observability, as shown in Fig 1

Many studies adopted the innovation concept in different fields and the context in light of 'relative advantage', observability, trialability, and complexity from the perspective of DOI theory to explain novel idea/technology diffusion and the related changes in behaviors (acceptance/rejection). Such characteristics affect new technology adoption depending on product-specific features [89].

### 7.2 Technology, organization and environment framework

[90] propose an analytical method known as the TOE framework that has since become the most popular acceptance technology theory that underpins IS studies and end-user adoption at organizations level [71]. TOE is used to conduct a structured analysis of innovation in organizations and to differentiate between intrinsic innovation, organizational capabilities, and motivations, as well as extensive environmental contexts affecting the users [91,92]. TOE involves different contexts (technological, organizational, and environmental). Fig 2 presents the TOE framework developed by Tornatzky and Fleischer and their influence on decision-making regarding technological innovation.

### 7.3 Theoretical framework and research hypotheses

There are various information system (IS) theories/models developed to study the acceptance of new technology. This study uses TOE and DOI theories, which comprise of Technological

**Table 2. The difference of CSF effect.**

| Author (Year) | Significant factor | Non-significant factor | Comments |
|---|---|---|---|
| [74] | Competitive pressure | | Firm size is eCRM adoption antecedents within the organizational context |
| [75] | Compatibility | Complexity | Firm size is significantly positively related to MHRS adoption. |
| [76] | | Complexity | |
| [66] | | Compatibility, security | Firm size does not play any role in the continuance of website adoption. |
| [77] | | Complexity | the adoption of Halal transportation. |
| [56] | Compatibility, top management support | Complexity, relative advantage, competitive pressure | Firm size is not significant. |
| [68] | Compatibility, competitive pressure | Relative advantages | Firm size is significantly positively related. |
| [78] | Relative advantages, compatibility | | Firm size is significantly positively related. |
| [69] | Relative advantage, compatibility, size, top management support | Competitive pressure | Firm size is significantly positively related to cloud computing adoption. |
| [75] | Relative advantage | | The size of the organization is positively related to the adoption of CRMS. |
| [79] | Product category | | Firm size is a significant factor in influencing CRM adoption. |
| [80] | Top management support | Customer pressure | |
| [81] | Compatibility, relative advantage, competitive pressure | | The size of the organization significantly influences cloud computing adoption in the MSMEs. |
| [82] | Owner's knowledge of IT | | while controlling for differences in firm attributes such as size. |
| [83] | | Competition | |
| [84] | Relative advantage | Competitive pressure | |
| [85] | CRM adoption | | The size of the firm is a significant determinant of adoption. |
| [77] | Customer pressure competitive pressure | Complexity | |
| [62] | Organizational characteristics have the most influence on adoption, followed by a set of environmental factors | Technology characteristics are not relevant to Malaysian companies | The size of the firm is a topic investigated in many types of research works regarding innovations and is noted as an indicator of organizational complexity, although some studies show a negative relationship between size and innovations. |
| [86] | | IT Infrastructure | The effects of its infrastructure on superior CRM capability are mediated through the capabilities of human analytics and business architecture |

context, Organizational Context, and Environmental context. The proposed framework is depicted in Fig 3.

# 8. Factors influencing CRM adoption

## 8.1 Technological context

Technology, Organization, Environment (TOE) framework was proposed by [90] to conduct a structured analysis of innovation in organizations and to differentiate between intrinsic

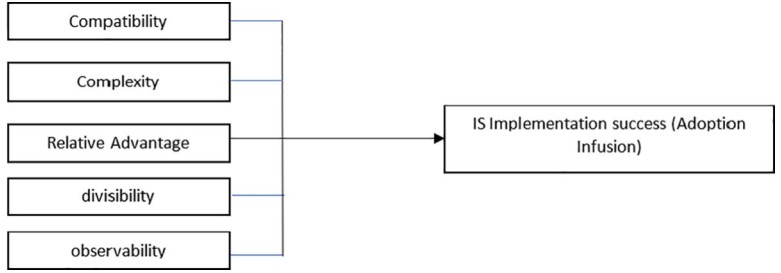

**Fig 1. Diffusion of innovation theory by [88].**

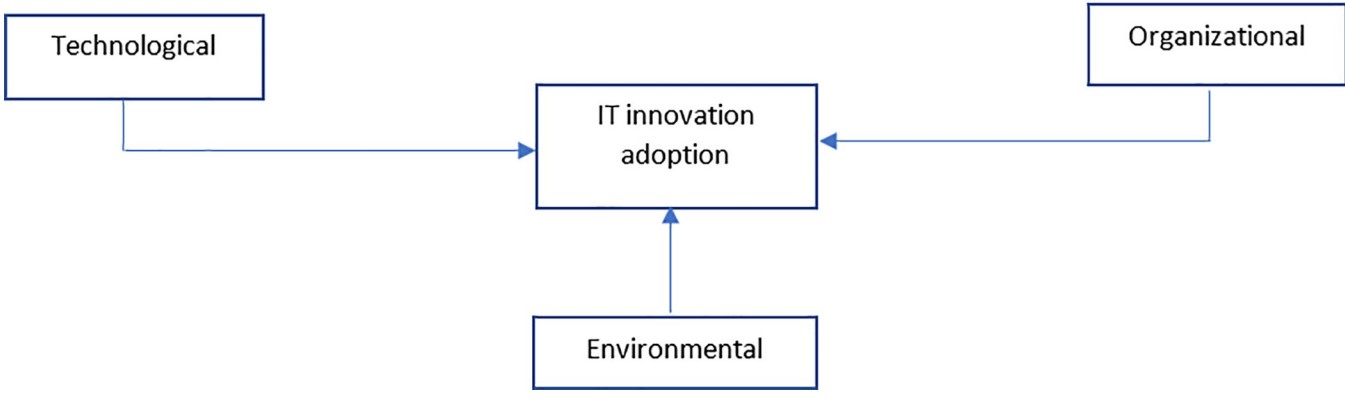

**Fig 2. Technology, organization and environment framework by [90].**

innovation, organizational capabilities, and motivations, as well as extensive environmental context affecting the users [91].

A technological context is important in enhancing the productivity of the organization [93,94]. Technological factors were examined using [85] diffusion theory, which asserts that adoption is affected by perceptions of five attributes of innovation, namely, 'relative advantage', complexity, compatibility, trialability, and observability.

In the context of small and medium-sized enterprises (SMEs), the rapid advancements in technology should take into consideration. SMEs have faced a considerable challenge in their attempt to keep abreast of technological innovations as their survival depends on information system (IS) use [93].

**i. Compatibility.** According to the DOI Theory, compatibility is a crucial technological feature perceived by the users. Compatibility drives the new system adoption decision. Several studies have been conducted to provide a description of the compatibility role and its determination of IT innovation adoption [57–62,66–79,81,82,85,95]. In relation to this, [69] reveal that innovation compatibility has a significant influence on the IS adoption among SMEs. Moreover, studies show that compatibility is significantly affected by CRM adoption and use [77,78,80].

[95] explains the importance of data integration to CRM adoption success. This indicates that CRM systems do not create the entire customer data alone but instead require other systems to convert data before feeding them to the CRM system. It is, therefore, crucial to ensure that the new CRM software is compatibility with the present operating systems in SMEs. In other words, compatibility between new technology and existing technology affects the firm's adoption process [96]. Compatibility reflects the level to which CRM has aligned with the organizations past experiences and current needs [62].

The new perspective to be examined in this regard would be how multinational corporations address global CRM compatibility. This study is a cross-sectional snapshot of a dynamic and relatively new phenomenon, and thus, it contributes merely a single data point in shedding light on the use of CRM and its adoption [97]. Thus, in this study, the following hypothesis is proposed:

**H1: Firm size moderates the relationship between compatibility and CRM adoption**

**ii. IT infrastructure.** 'IT infrastructure' brings about the development of major applications, 'information sharing' across products/services, and the implementation of transaction processing as well as inter-organizational systems [98]. It also plays a key role in reinforcing the capabilities of human analytic and business architecture [86].

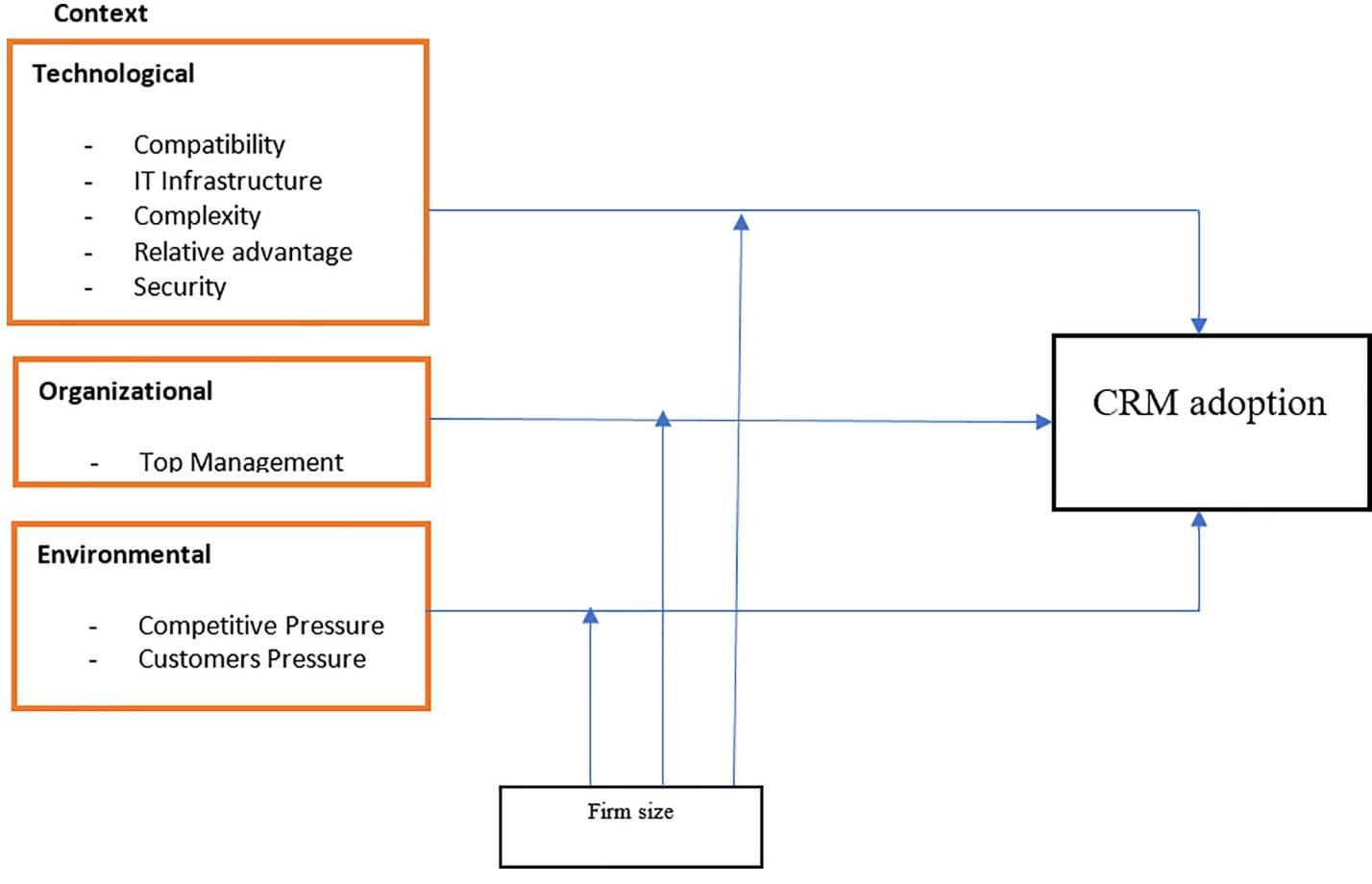

**Fig 3. Proposed conceptual framework for CRM adoption.**

Several studies emphasized that 'IT infrastructure' has increasingly become the core of business operations such as CRM [57] and [87,88]. CRM literature has dedicated the importance of 'IT infrastructure', staff knowledge and skills for effective utilization of IT resources, interaction, and manage customers information [89,90]. In other words, CRM investment calls for sufficient 'IT infrastructure' that can be justified in light of cost savings and generation of profitability [99]. More importantly, a value chain includes technological resources, 'IT infrastructure', and CRM processes can lead to organizational performance [17,65].

Moreover, Developments in hardware and software have equipped firms with various solution alternatives to support CRM. The business value that may not have been possible in the technology itself but rather from the capability of obtaining information from the entire customer touch-points, telesales, service departments, direct sales and channel partners [86]. New software system adoption may call for the enhancement or the addition of 'IT infrastructure' to install new CRM software [5]. CRM failures can be avoided if the CRM strategies are linked with the IT infrastructure and the firm's stakeholders (employees, customers, channels) infrastructure [100]. Thus, this study proposes the following hypothesis:

**H2: Firm size moderates the relationship between 'IT infrastructure' and CRM adoption**

**iii. Complexity.** Although CRM has been viewed as a pure technology a mistake, IT has a key role in successful CRM projects, and CRM technologies complexity positively correlates

with the company's level of advancement [101]. There are many reasons related to failed CRM objectives, with some being inevitable, while others can be clear [99]. In general, failed CRM projects are caused by complex technical and organizational issues that are related to system implementation [102].

CRM failure has been related in the literature to various CRM components and processes. For instance, the system complexity leads to differences among the tactical stress of stakeholders and the lack of consensus concerning the accurate CRM definition [103]. When considering the CRM adoption complexity and diversity, the proposed conceptual framework focuses on providing common requirements and conditions for CRM adoption among businesses.

The majority of related studies reveal a significant relationship between complexity and different fields such as e-commerce adoption [104], electronic record management [105], cloud computing [69], halal transportation services [77] and CRM adoption [106]. Contrasting to other characteristics of innovation, complexity negatively linked with the adoption probability [69]. Although complexity may not be significant as 'relative advantage', it has been evidenced to prevent the adoption of technology [87]. The study proposes the following hypothesis:

**H3: Firm size moderates the relationship between complexity and CRM adoption**

**iv. Relative advantage.** 'Relative advantage' is among the five innovation characteristics that significantly impact the IT innovation adoption among firms [26,57–62,66–91,93,95–98,107–109]. [75] reveal that the 'relative advantage' significantly affected the adoption of CRM systems in hospitals. Moreover, [58] and [61,62,66–79,81,82,85,95] consider a 'relative advantage', complexity, and compatibility as the most significant technological factors in determining the adoption of CRM.

Past research has examined 'relative advantage' and found it to be a major variable in innovation adoption [69,110]. The variable is one of the DOI factors which has been revealed to be a positive influence in the innovation adoption [62].

The 'relative advantage' of technology over another is a major antecedent of new technology adoption, and it plays a key role in innovation adoption [90]. In a related study, [111] combined TAM with IDT; their empirical findings indicated several guidelines to enhance CRM diffusion among organizations. Their findings also showed that complexity and compatibility influence CRM adoption indirectly via 'relative advantage', as a mediating variable, they argue that the 'relative advantage' has a key role being a predictor of CRM adoption among employees.

Even though the top predictors in the literature include perceived usefulness, 'relative advantage', cost, security, compatibility, complexity, and observability. The two technological and innovation characteristics that are significant in the level of organizations are 'relative advantage' and compatibility [66]. In a related study, [62] defines 'relative advantage' as the level to which the CRM technology develops customer information that could enable them to provide optimum customer services when compared to traditional methods to meet customer needs. Therefore, the study proposes the following hypothesis;

**H4: Firm size moderates the relationship between 'relative advantage' and CRM adoption**

**v. Security.** Security, another technological characteristic, is a significant determinant of the adoption of IT within organizations [112]. This factor was also left out from the existing frameworks despite its significance to the organizations' performance and plays a key role in supporting CRM implementation and adoption [113,114].

Moreover, [115] revealed that security threat to be the top critical adoption barrier in various fields such as health information systems [116] and electronic commerce [66]. However, security was not found to be a major barrier to adopting a decision support system as reported

by [117], security seemed to be a concern only when problems occurred. Therefore, the following hypothesis is proposed:

**H5: Firm size moderates the relationship between security and CRM adoption**

### 8.2 Organizational context

Organizational characteristics refer to parameters influencing the adoption decision of the innovation, and it reflects the highest effect on CRM adoption [62]. There are many theories and models proposed in IS disciplines that address technological, environmental, and organizational characteristics as robust predictors of technology adoption in different contexts [102,103]. For instance, halal warehouse adoption [118], CRM implementation [119], telecommunication [62] and human resources information systems [120].

In a related study, successful innovation implementation was found to be significantly related to the organization and environmental contexts [111]. Meanwhile, [121] argues that the main reason for the failure of CRM projects is considering CRM as a technological tool and not assuming the various organizational and cultural changes involved.

**i. Top management support.** Top management recognizes that customers are the core of businesses, and the success of a company relies on effective relationships management with customers. In order to meet customers' expectations, companies should adopt oriented marketing strategies to a high-quality relationship with customers to ensure the company's success [122].

Support from top management has a key role in the adoption of cloud computing, specifically in guiding resource allocation, services integration, and re-engineering of processes [123]. In other words, 'top management support' is an important factor that influences the IS function effectiveness in the organization [124]. [125] evidenced that top management as a factor that enables the sharing of knowledge among the members of the organization.

The adoption of CRM requires 'top management support'. Successful CRM adoption is effective when top management is committed to change; hence implement the CRM initiative [26,53–62,66–93,95–115] and [26,78,80,83,84,86–93,96–117]. 'Top management support' has been the topmost significant determinant of the maintenance of successful structural transformation and attitudes changes among employees to adopt CRM [126]. This is supported by [127] and [13], who contended that top management is responsible for creating an alignment between the new CRM system and the present business practices. Perhaps it is a critical factor to success in the adoption of the new technology [56]. [6,10,11,13–25,27–32,34–41,43–53,63–65] asset that the lack of top management commitments becomes the main barrier to the success of the endeavor. Hence, the following hypothesis is suggested:

**H6: 'Firm size moderates the relationship between 'top management support' and CRM adoption**

### 8.3 Environmental context

Environmental context is linked to the operational facilitators and inhibitors, with the significant among them are competitive pressure, the readiness of trading partners, government encouragement, and technological [71]. Even though DOI has been extensively employed by studies to examine the adoption of new technology at the organizational level, it does not include environmental issues [128]. The present study adopts the environmental context as a study variable, and thus, it uses the TOE framework rather than the DOI, as suggested by [129]. The TOE framework addresses the environmental context in predicting intra-firm innovation technology adoption.

**i. Customer pressure.** The firm has to be aware of the needs and demands of the customers. If a potential customer controls the relationship, this would mean the future of the firm

business is in the customer's hands. If the customer is satisfied (with little pressure), then the firm will likely consider its systems as being at a satisfactory level. Thus, there is an inverse relationship between perceived customer pressure and overall satisfaction [130].

[108] described 'customer pressure' as the demands and behaviors of customers that make companies adopt new technologies. It is the requests and requirements of primary stakeholders (end consumers and business consumers) regarding the organizations environmental and social performance to be improved [131]. 'Customer pressure' is the focal firm's motivation to adopt ISO standards; hence, it has a positive relationship with quality control performance [132]. Thus, there is a significant and positive relationship between 'customer pressure' and the adoption intention of SMEs [133]. This is proven when [131] revealed that 'customer pressure' is a top determinant of the Firm's Environmental Performance. It even has a significant effect on green innovation adoption among SMEs [92,104–106,111,112,115–124,134]. Hence, the next hypothesis of this study is as follows:

**H7: Firm size moderates the relationship between 'customer pressure' and CRM adoption.**

**ii. Competitive pressure.** Globalization, growth technology, and 'competitive pressures' lead organization leaders to adopt a suitable competitive strategy to achieve maximum market share [125,135]. A competitive market environment enhances innovation among organizations [136]. [84] explained that empirical studies have revealed that higher innovation adoption likelihood is linked to higher 'competitive pressure'. In particular, [57] examined e-records management system adoption in higher professional education institutions and concluded that "the adopters are beneath higher 'competitive pressure' than the non-adopters." A review of e-commerce adoption among SMEs in Malaysia revealed that adopters of e-commerce had a higher tendency to adopt innovative system In a highly competitive environment [64,84].

'Competitive pressure' is one of the main reasons for investing in CRM, and this is particularly true if CRM is viewed as an asset that allows organizations to increasingly focus on their customers [26,67–93,95–127,134,135]. Competitive put pressure on the organization to be more creative and forces its leaders to adopt strategies to increase market share [135].

When a rival company employs a CRM system, other companies of the same caliber have the urgency to adopt such a system as well. Hence, both 'customer pressure' and 'competitive pressure' are the top predictors of social CRM adoption [107]. In the Arab world case, particularly Saudi Arabia, 'competitive pressure' drives business organizations to keep track of their customers and identify their needs to provide service customization [137]. Thus, this study proposes the following hypothesis:

**H8: Firm size moderates the relationship between 'competitive pressure' and CRM adoption**

## 9. Method

### i. Data collection procedure and sample data

Data for this study was gathered from employees working in SMEs in Palestine. This includes general managers, heads of department, operational employees' technology who are involved in activities related to CRM technology, and the adoption decision. The e-version of the questionnaire was uploaded to the survey website, coupled with an introduction page that provided the definitions of CRM and SMEs. An invitation letter was then followed, and the survey link to the SME employees.

From a total of 420 respondents who received the link, 331 (79%) filled and returned the survey. This response rate was achieved after countless efforts to encourage the participation

of respondents in the study. According to [43], a response rate of 50% or more is sufficient and acceptable for analysis, 60% and over is good, and over 70% is excellent. Thus, the rate of response to this internet survey is acceptable in this study.

## ii. Sample profile

The demographic data collected in this study are such as age, gender, educational level, job level, department, and organizational tenure. This is suggested even if the theoretical framework does not need the variables as the obtained data will shed insight into the sample characteristics in the report following data analysis [138]. The demographic variables and other background information are shown in Table 3.

The actual study had 323 usable questionnaires, and the data obtained were exposed to confirmatory factor analysis and model testing. As shown in Table 5.4, the majority of respondents belong to the male group, with a percentage of 81.1%, followed by the female group (18.9%). The majority of respondents belong to a bachelor group with a percentage of 57%, followed by the postgraduate group, with a rate of 34%. The diploma group has a percentage of 9%. Respondent distribution and precedence of bachelor, postgraduate, diploma are normal in Palestine society. This study considers four main company types. The highest percentage of respondents are in the Sales company (29.4%), followed by respondents working in ICT companies (28.8%), then the service company (27.6 and lastly, product company 14.2%. The majority of the respondents are heads of department (46.7%), several are general managers (31.1%), while the rest are operational employees (22%).

Since this study is focused on SMEs, where the number of employees ranges from 5 to 49, companies that have less than five or over 49 employees were excluded. Based on the number of employees, 60.7% of the respondents are working in SMEs that have 5–19 employees, while 39.3% of the respondents are working in companies that have 20–49 employees. The work

**Table 3. Summary of the demographic profile.**

| | Description | Frequency | Percentage |
|---|---|---|---|
| Gender | Male | 262 | 81.1 |
| | Female | 61 | 18.9 |
| Level of education | Diploma | 29 | 9.0 |
| | Bachelor's Degree | 184 | 57 |
| | Master's Degree | 83 | 25.6 |
| | Doctorate Degree | 27 | 8.4 |
| Type of company | Information and Communications Technology (ICT) | 93 | 28.8 |
| | Product company | 46 | 14.2 |
| | Service company | 89 | 27.6 |
| | Sales company | 95 | 29.4 |
| Position in organization | General manager | 101 | 31.3 |
| | Head of the department | 151 | 46.7 |
| | Operational employees | 71 | 22 |
| Number of employees in an organization | 5–19 | 196 | 60.7 |
| | 20–49 | 127 | 39.3 |
| Years of experience | <1 year | 6 | 1.9 |
| | 1–5 years | 70 | 21.6 |
| | 5–10 years | 49 | 15.2 |
| | >10 years | 198 | 61.3 |
| Age | 20–30 years old | 63 | 19.5 |
| | 31–40 years old | 97 | 30 |
| | 41–50 years old | 63 | 19.5 |
| | 51 and above | 100 | 31 |

experience of the respondents is classified into three groups, respondents with less than a year of experience constitute 1.9%, those with 1–5 years constitute 21.6%, those with 5–10 years constitute 15.2%, and employees with ten years of experience and over constitute 61.3%. The majority of the respondents are in the age group of 51 and above (31%) followed by the age group of 31–40 (30%), then the age group 41–50 (19.5%), while the rest are 20-30years old (19.5%).

## 10. Analysis of data and presenting results

### I. Testing of the moderation effects of firm size

A moderator variable is one that affects the relationship between two variables. In other words, the impact of the independent variable on the dependent variable varies according to the level of the moderator [139]. In this study, SEM was used to examine the moderation effects of firm size on the impact of compatibility (CMP), complexity (CMX), competitive pressure (COP), customers pressure (CUP), relative advantage (RLA), security (SEC), top management support (TMS) and IT infrastructure (ITI) as independent variables on CRM adoption as the dependent variable.

Table 4 shows that the firm size has no statistical significance on competitive pressure, security, complexity, and relative advantage and CRM adoption This result thus fails to support H3, H4, H5 and H8. While, TMS, CMP, CUP, and ITI had a significant effect on CRM adoption as their p-values were all lower than the standard significance level of 0.05.

## 11. Findings and discussion

This study primarily aims to examine the moderating effects of the firm size in the adoption of CRM in Palestinian SMEs. Categorized into technological, organizational, and environmental contexts. The study findings provided descriptions of the exogenous variables-endogenous variables relationship, using TOE framework, DOI theories. Table 5 lists the tested hypotheses and the results.

In a technological context, the findings show that both variables' compatibility and IT infrastructure are related with the intention to adopt CRM. As for compatibility, the moderation effect of firm size has a positive indirect impact, while IT infrastructure has a negative on behavioral intention to use CRM technology as shown in Figs 4 and 5. This empirical finding is aligned with the findings reported by other scholars who found the significant role of compatibility and IT infrastructure on the adoption of technology [67].

In an organizational context, the findings indicate that firm size moderates the positive effect of top management support on CRM adoption, as shown in Fig 6. In any business, top management plays a vital role in the adoption process as it is the one that decides for the best

**Table 4. Moderation effects of firm size.**

|  | Path | Path Coefficient (β) | t Statistics | p-Value | Hypothesis Result |
|---|---|---|---|---|---|
| H1 | CMP * FS → CAD | 0.049 | 1.805 | **0.036** | Supported |
| H2 | ITI * FS → CAD | -0.058 | 2.030 | **0.021** | Supported |
| H3 | CMX * FS → CAD | 0.044 | 1.155 | 0.124 | unsupported |
| H4 | RLA * FS → CAD | 0.008 | 0.308 | 0.379 | unsupported |
| H5 | SEC * FS→ CAD | -0.011 | 0.275 | 0.392 | unsupported |
| H6 | TMS * FS → CAD | 0.048 | 1.703 | **0.045** | Supported |
| H7 | CUP * FS → CAD | -0.066 | 2.350 | **0.010** | Supported |
| H8 | COP * FS →CAD | -0.005 | 0.152 | 0.440 | unsupported |

**Table 5. Summary of the tested hypothesis.**

| | Hypothesis (firm size as a moderator) | Results are consistent with previous researches finding |
|---|---|---|
| H1 | Compatibility→ CRM adoption | The hypothesis is supported. These results are in line with [56,67] |
| H2 | IT infrastructure → CRM adoption | The hypothesis is supported. These results are in line with [86] |
| H3 | Complexity→ CRM adoption | The hypothesis is not supported. These results are in line with [75] |
| H4 | Relative advantage → CRM adoption | The hypothesis is not supported. These results are in line with [75] |
| H5 | Security → CRM adoption | The hypothesis is not supported. These results are in line with [66] |
| H6 | Top management support → CRM adoption | The hypothesis is supported. These results are in line with [57,71] |
| H7 | Customers pressure → CRM adoption | The hypothesis is supported. These results are in line with [80] |
| H8 | Competitive pressure→ CRM adoption | The hypothesis is not supported. These results are in line with [69] |

of the organization [140]. 'Top management support' is the topmost significant determinant in the maintenance of successful structural transformation and attitudes changes among employees with more skills to adopt CRM [126].

In an environmental context, the findings indicate that firm size moderates the negative effect of customer pressure intention toward CRM adoption, as shown in Fig 7. Empirical results show that competitive pressure is a top driver for the adoption and diffusion of IT [141]. Customer pressure urges firms towards adopting new technologies [80]. This result is aligned with another previous study, such as carried out by [80], who revealed the positive impact of customer pressure on the intention of e-commerce technology adoption in Saudia Arabia. Customer pressure was ranked as the third most determinant of adoption of e-customer relationship management among 12 variables by [142] study.

Although firm size significantly moderates the relationship between ITI and CRM adoption, the direct effect of IT Infrastructure was not significant. This result is in line with prior research [86], who stated that there is indirect-only mediation and implies that although IT Infrastructure does not have a significant direct effect on Superior CRM, it does have a strong

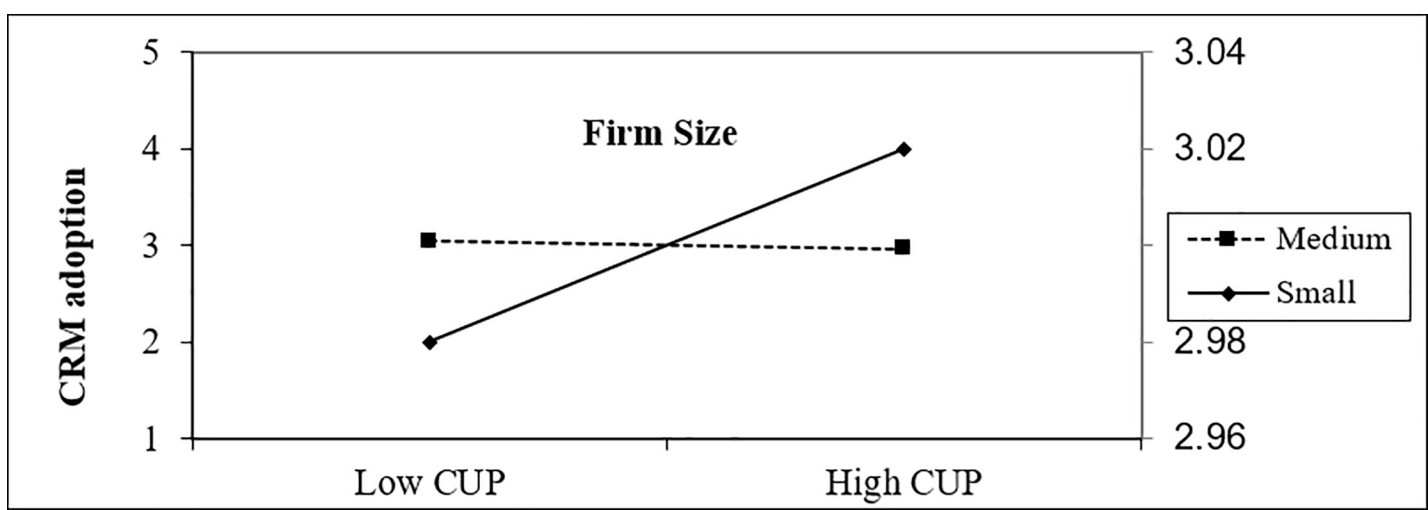

**Fig 4. Moderating effect of firm size on the impact of compatibility on CRM adoption.**

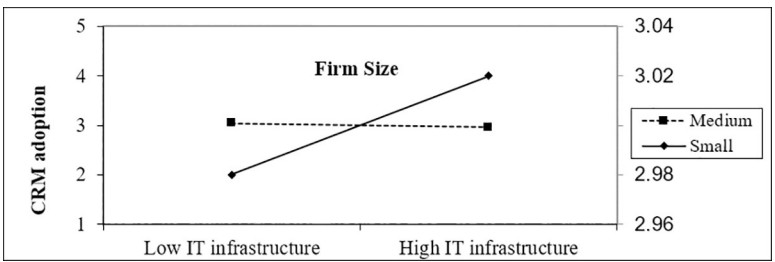

**Fig 5. Moderating effect of firm size on the impact of IT Infrastructure on CRM adoption.**

indirect effect. The following addresses the results of the path analysis of the moderation effects of firm size in relation to the hypotheses (H1, H2, H3, H4, H5, H6, H7, and H8) in the structural model:

i. H1: firm size tools moderate the relationship between compatibility and CRM adoption ($\beta$ = -0.049, t = 1.805, p-value = 0.036). Thus, H1 was supported.

ii. H2: firm size tools moderate the relationship between IT infrastructure and CRM adoption ($\beta$ = -0.058, t = 2.030, p = 0.021). Thus, H2 was supported.

iii. H3: firm size tools did not moderate the relationship between complexity and CRM adoption ($\beta$ = 0.044, t = 1.155, p = 0.124). Thus, H3 was not supported.

iv. H4: firm size tools did not moderate the relationship between relative advantage and CRM adoption ($\beta$ = 0.008, t = 0.308, p = 0.0379). Thus, H4 was not supported.

v. H5: firm size tools did not moderate the relationship between security and CRM adoption ($\beta$ = -0.011, t = 0.275, p = 0.392). Thus, H5 was not supported.

vi. H6: firm size tools moderate the relationship between top management support and CRM adoption ($\beta$ = 0.048, t = 1.703, p-value = 0.045). Thus, H6 was supported.

vii. H7: firm size tools moderate the relationship between customers pressure and CRM adoption ($\beta$ = -0.066, t = 2.350, p = 0.010). Thus, H7 was supported.

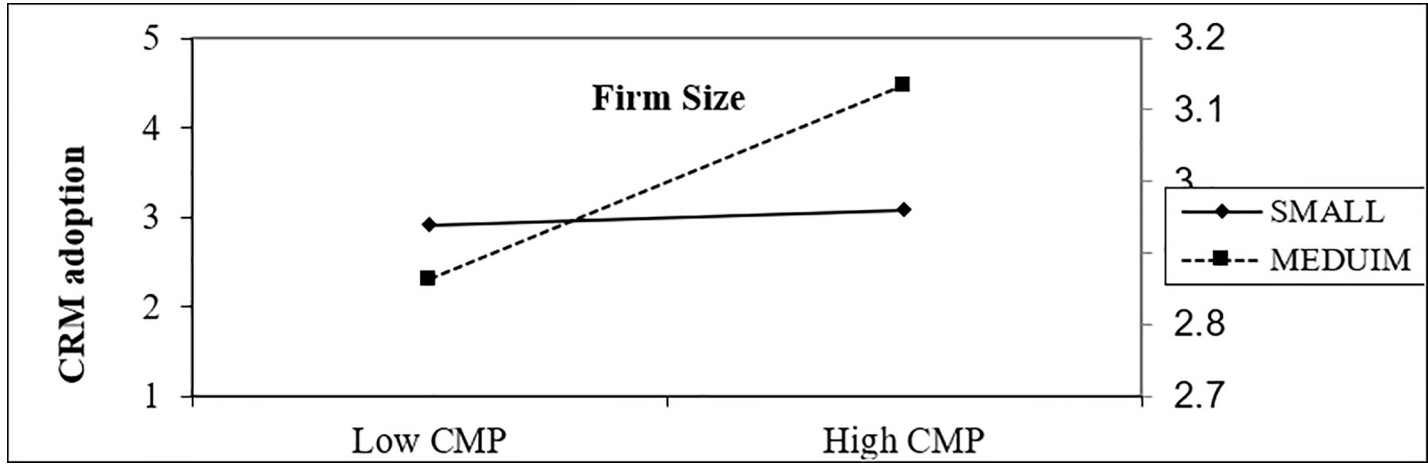

**Fig 6. Moderating effect of firm size on the impact of customer pressure on CRM adoption.**

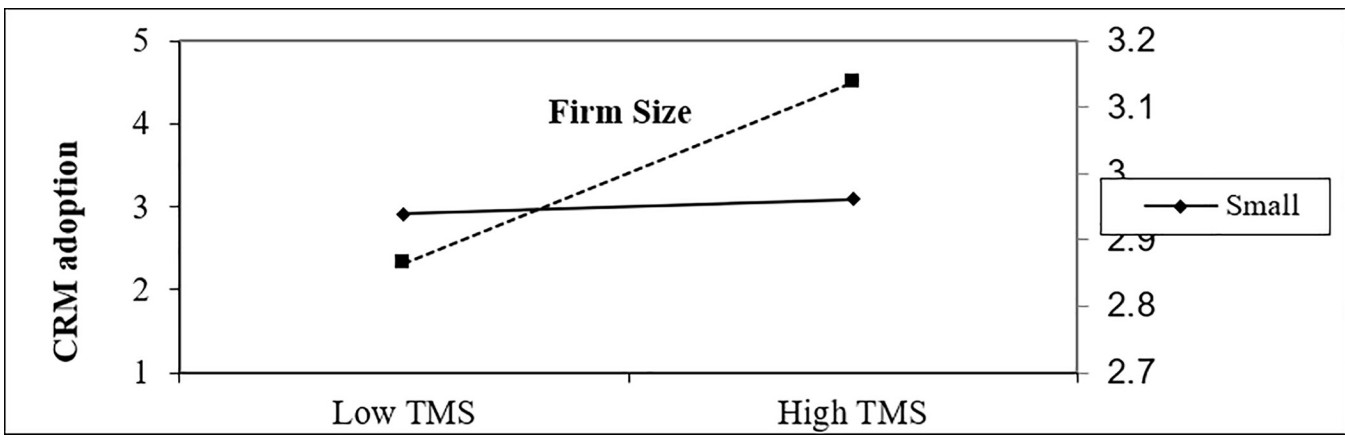

**Fig 7. Moderating effect of firm size on the impact of top management support on CRM adoption.**

viii. H8: firm size tools did not moderate the relationship between competitive pressure and CRM adoption ($\beta$ = -0.005, t = 0.152, p-value = 0.440). Thus, H8 was not supported. Fig 8 displays the structural model for testing the moderation firm size generated by SMART-PLS.

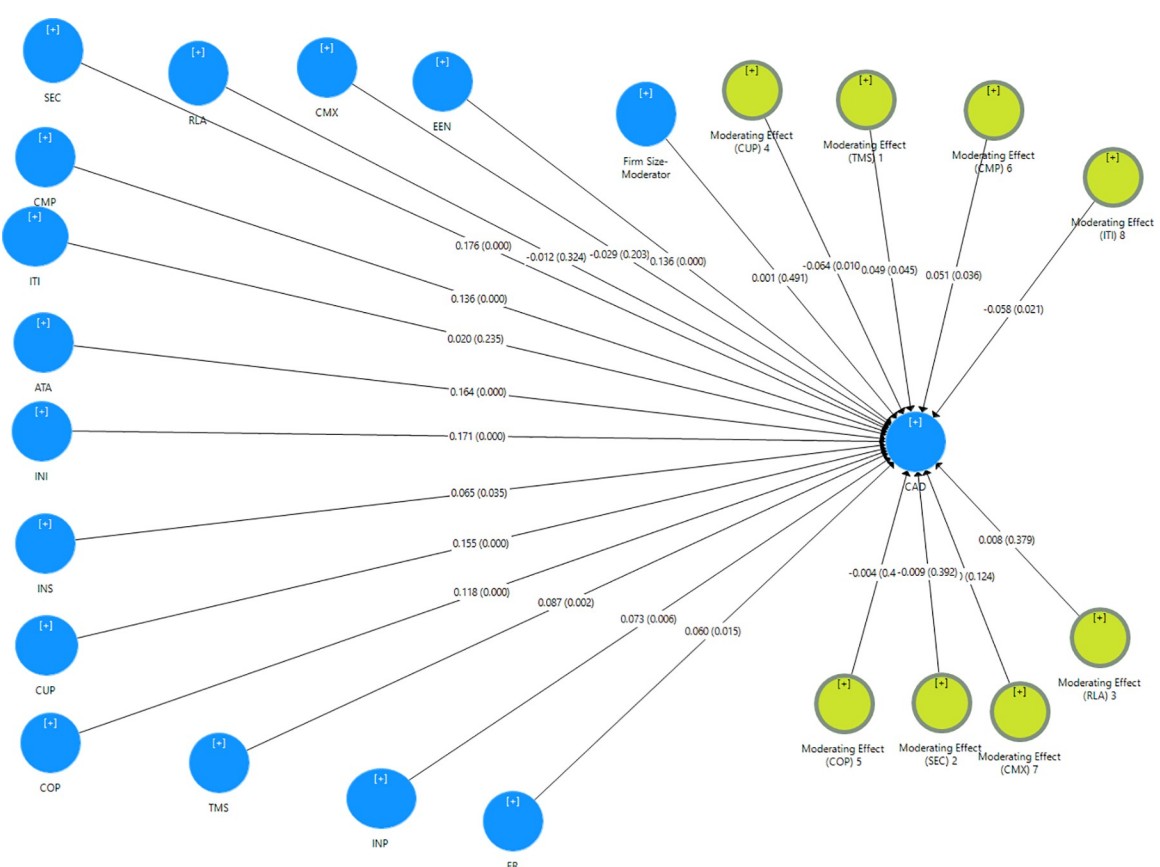

**Fig 8. The structural model for testing the moderation firm size.**

To conclude, the new proposed framework would assist SMEs in Palestine to embark on the appropriate and complete CRM initiative. SMEs should use this framework to understand customer value, which could supersede the quality of the product. Thus, the concern of companies has shifted to meeting customer satisfaction, the attraction of potential customers, and maintaining present customers' loyalties through CRM establishment [9,10,11,13–21,23–25,27–32,63].

## 12. Conclusion

Companies, where CRM is incorporated as a business strategy, tend to grow faster than those which did not. This is due to the objective of the initiative whereby enhancing the customers' relationships, which in turn leads to maximized revenue, optimized profit, improved productivity, and customer satisfaction become the ultimate aim. CRM also has the capability of integrating the entire company's marketing efforts and automates specific customer-organization relationships. In this modern environment where business operations are dependable on the advances of technology, companies should be striving to adopt a system such as CRM in order to be more effective and efficient. As a system, CRM has many untold advantages and should be considered as worthy of investment in the long run.

This study attempts to examine the moderator effect of firm size on the effects of compatibility (CMP), complexity (CMX), 'competitive pressure' (COP), 'customers pressure' (CUP), 'relative advantage' (RLA), security (SEC), 'top management support' (TMS) and 'IT infrastructure' (ITI) as independent variables on CRM adoption as the dependent variable. The findings indicate that firm size moderate the relationship between top management support, compatibility, customer pressure, IT infrastructure, and CRM adoption. While firm size did not moderate the relationship between competitive pressure, security, complexity, and relative advantage and CRM adoption. Moreover, this study contributes to both academics and business practitioners by providing insights into factors moderation by firm size that affect CRM adoption in Palestinian SMEs, which has never been explored before. The findings of this study are limited to generalization toward Palestinian SMEs, and those neighboring countries similar to Palestine in culture and situation. The study, however, fell short of covering all the SME industry groups in Palestine. Thus, future studies may examine the effect of CRM technology in different industries, sectors, and economies.

### Ethical approval

This study has been approved by UKM (Universiti Kebangsaan Malaysia) Ethics Committee. If needed, verification of approval can be obtained either by writing to professor Zawiyah or Dr Hazura Mohamed. The date on which the study was carried out in 12/2019

### Supporting information

**S1 File.**
(SAV)

**S2 File.**
(DOCX)

### Acknowledgments

The study is financially supported by Center for Software Technology and Management, Faculty of Information Science and Technology, Universiti Kebangsaan Malaysia and Palestine Technical University–Kadoorie.

## Author Contributions

**Conceptualization:** Omar Hasan Salah, Zawiyah Mohammad Yusof, Hazura Mohamed.

**Supervision:** Hazura Mohamed.

**Validation:** Hazura Mohamed.

**Visualization:** Hazura Mohamed.

**Writing – original draft:** Omar Hasan Salah.

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
