## [Decision Letter · Decision Letter 0]

10 Feb 2020

PONE-D-19-33183

The Determinant Factors for the Adoption of CRM in the Palestinian SMEs: The Moderating Effect of Firm Size

PLOS ONE

Dear Author(s),

Thank you for submitting your manuscript to PLOS ONE. After careful consideration, we feel that it has merit but does not fully meet PLOS ONE’s publication criteria as it currently stands. Therefore, we invite you to submit a revised version of the manuscript that addresses the points raised during the review process.

We would appreciate receiving your revised manuscript by Mar 26 2020 11:59PM. To enhance the reproducibility of your results, we recommend that if applicable you deposit your laboratory protocols in protocols.io, where a protocol can be assigned its own identifier (DOI) such that it can be cited independently in the future. For instructions see: http://journals.plos.org/plosone/s/submission-guidelines#loc-laboratory-protocols

We look forward to receiving your revised manuscript.

Kind regards,

Amira M. Idrees, Associate Professor

Academic Editor

PLOS ONE

Journal Requirements:

Reviewers' comments:

Reviewer's Responses to Questions

**Comments to the Author**

1. Is the manuscript technically sound, and do the data support the conclusions?

Reviewer #1: Partly

Reviewer #2: Partly

2. Has the statistical analysis been performed appropriately and rigorously? 

Reviewer #1: N/A

Reviewer #2: N/A

3. Have the authors made all data underlying the findings in their manuscript fully available?

Reviewer #1: No

Reviewer #2: No

4. Is the manuscript presented in an intelligible fashion and written in standard English?

Reviewer #1: Yes

Reviewer #2: No

5. Review Comments to the Author

Reviewer #1: Please refer to the attached referee report for the detail comments on the submitted paper pertaining to the overall writing, the discussions on the methods/findings, and all the other comments about the paper.

Reviewer #2: There are many reasons why the studies on the adoption of CRM are interesting and, therefore, it is relevant that contributions be made in this field. In the paper a series of contributions have been made in this regard, and especially in the analysis the moderating effects of the firm size in the adoption of CRM. However, for the publication of this paper I recommend making the following modifications:

The English level of the article is standard, but could be improved by a native translator with knowledge in economic scientific literature.

The paper does not follow the basic structure of the work in PLOS ONE: introduction, materials and methods, results, discussion and conclusions (optional). This is especially serious in the Analysis of data and presenting results section, in which authors are cited in the presentation of results. The discussion should appear in your particular section and not integrated with the results.

In method, it is not specified when the study has been carried out, at least clearly. Likewise, it is necessary to provide data on the degree of viability of the proposed model and information on its adjustment.

6. PLOS authors have the option to publish the peer review history of their article (what does this mean?). If published, this will include your full peer review and any attached files.

Reviewer #1: No

Reviewer #2: No

---

## [Author Response · Author response to Decision Letter 0]

25 Jun 2020

I had attached a file to specific reviewer and editor comments in "Comments" file

---

## [Decision Letter · Decision Letter 1]

16 Sep 2020

PONE-D-19-33183R1

The Determinant Factors for the Adoption of CRM in the Palestinian SMEs: The Moderating Effect of Firm Size

PLOS ONE

Dear Author

Thank you for submitting your manuscript to PLOS ONE. After careful consideration, we feel that it has merit but does not fully meet PLOS ONE’s publication criteria as it currently stands. Therefore, we invite you to submit a revised version of the manuscript that addresses the points raised during the review process.

We look forward to receiving your revised manuscript.

Kind regards,

Amira M. Idrees, Associate Professor

Academic Editor

PLOS ONE

Reviewers' comments:

Reviewer's Responses to Questions

**Comments to the Author**

1. If the authors have adequately addressed your comments raised in a previous round of review and you feel that this manuscript is now acceptable for publication, you may indicate that here to bypass the “Comments to the Author” section, enter your conflict of interest statement in the “Confidential to Editor” section, and submit your "Accept" recommendation.

Reviewer #2: (No Response)

2. Is the manuscript technically sound, and do the data support the conclusions?

Reviewer #2: Yes

3. Has the statistical analysis been performed appropriately and rigorously? 

Reviewer #2: I Don't Know

4. Have the authors made all data underlying the findings in their manuscript fully available?

Reviewer #2: Yes

5. Is the manuscript presented in an intelligible fashion and written in standard English?

Reviewer #2: Yes

6. Review Comments to the Author

Reviewer #2: Reviewer’s report: Studies on CRM are very important because their implementation incompanies digitally systematize the commercial function of companies,which in this complex stage derived from COVID-19, can be very relevantfor the management of companies. However, for publication I recommendmaking the following modifications:Abstract:The authors must clearly specify the objective of the study at thebeginning of the study, and then expose the importance and need for it,without going into the definition of the concept of CRM within thisabstract. The results are presented but no mention is made of the finalconclusions.Introduction:The structure of the article must have four very clear sections:introduction, method, results and discussion and conclusion. Authorsshould make any necessary adjustments to provide this clarity.Material and methods- The name of the Ethics Committee that supervised the study must bestated.- The date on which the study was carried out is not specified, at leastclearly.- The methodology used (The Partial Least Square-Structural EquationModel (PLS-SEM)) is not justified, nor does it seem to be specified in thetext or it is not made clearly.- It is necessary to justify better because the case of Palestine is beingstudied.DiscussionThe discussion is pretty sparse. It is well linked with previous studies, butit is necessary to show a greater richness that more clearly shows thecontribution of this work.

- Personally, I have understood the entire exposition in English of the pa-per, but I am not a native English speaker. I consider it appropriate thatsomeone from your journal check the writing in this language.

7. PLOS authors have the option to publish the peer review history of their article (what does this mean?). If published, this will include your full peer review and any attached files.

Reviewer #2: No

---

## [Author Response · Author response to Decision Letter 1]

18 Sep 2020

Comments :1- Abstract: The authors must clearly specify the objective of the study at the beginning of the study, and then expose the importance and need for it, without going into the definition of the concept of CRM within this abstract. The results are presented but no mention is made of the final conclusions. Page 1

2- Introduction: The structure of the article must have four very clear sections: introduction, method, results and discussion and conclusion. Done

3- Authors should make any necessary adjustments to provide this clarity. Material and methods- The name of the Ethics Committee that supervised the study must be stated and The date on which the study was carried out is not specified, at least clearly. 21

4- The methodology used (The Partial Least Square-Structural EquationModel (PLS-SEM)) is not justified 7-8

5- It is necessary to justify better because the case of Palestine is being studied. 20

76-PLOS authors have the option to publish the peer review history of their article (what does this mean?). If published, this will include your full peer review and any attached files. Done

---

## [Decision Letter · Decision Letter 2]

21 Oct 2020

PONE-D-19-33183R2

The Determinant Factors for the Adoption of CRM in the Palestinian SMEs: The Moderating Effect of Firm Size

PLOS ONE

Dear Author(s)

Thank you for submitting your manuscript to PLOS ONE. After careful consideration, we feel that it has merit but does not fully meet PLOS ONE’s publication criteria as it currently stands. Therefore, we invite you to submit a revised version of the manuscript that addresses the points raised during the review process.

We look forward to receiving your revised manuscript.

Kind regards,

Amira M. Idrees, Associate Professor

Academic Editor

PLOS ONE

Reviewers' comments:

Reviewer's Responses to Questions

**Comments to the Author**

1. If the authors have adequately addressed your comments raised in a previous round of review and you feel that this manuscript is now acceptable for publication, you may indicate that here to bypass the “Comments to the Author” section, enter your conflict of interest statement in the “Confidential to Editor” section, and submit your "Accept" recommendation.

Reviewer #2: (No Response)

2. Is the manuscript technically sound, and do the data support the conclusions?

Reviewer #2: Yes

3. Has the statistical analysis been performed appropriately and rigorously? 

Reviewer #2: I Don't Know

4. Have the authors made all data underlying the findings in their manuscript fully available?

Reviewer #2: Yes

5. Is the manuscript presented in an intelligible fashion and written in standard English?

Reviewer #2: Yes

6. Review Comments to the Author

Reviewer #2: Dear Author,

Re: Manuscript “The Determinant Factors for the Adoption of CRM in the Palestinian SMEs: The Moderating Effect of Firm Size”

Reviewer’s report:

The authors have made significant improvements to the text. At the end of each section, the status of each of the aspects mentioned in the previous review is exposed:

Abstract:

The authors must clearly specify the objective of the study at the beginning of the study, and then expose the importance and need for it, without going into the definition of the concept of CRM within this abstract. The results are presented but no mention is made of the final conclusions: This aspect is resolved.

Introduction:

The structure of the article must have four very clear sections: introduction, method, results and discussion and conclusion. Authors should make any necessary adjustments to provide this clarity. In this regard, authors must place point 5. justifications for using PLS within the Method section.

Material and methods

- The name of the Ethics Committee that supervised the study must be stated. This aspect is resolved. The date of the study must appear in the Method section.

- The date on which the study was carried out is not specified, at least clearly. As stated in the previous section, the date of the study must appear in the Method section.

- The methodology used (The Partial Least Square-Structural Equation Model (PLS-SEM)) is not justified, nor does it seem to be specified in the text or it is not made clearly. The authors have presented the characteristics of this methodology, but it would still be necessary to adapt these characteristics to the case studied.

- It is necessary to justify better because the case of Palestine is being studied. It is understood that the authors explain this in point 3 3.

THE PALESTINE MARKET AND SMES.

Discussion

The discussion is pretty sparse. It is well linked with previous studies, but it is necessary to show a greater richness that more clearly shows the contribution of this work. This aspect is resolved with what is stated in the text.

- Personally, I have understood the entire exposition in English of the paper, but I am not a native English speaker. I consider it appropriate that someone from your journal check the writing in this language.

Best regards

7. PLOS authors have the option to publish the peer review history of their article (what does this mean?). If published, this will include your full peer review and any attached files.

Reviewer #2: No

---

## [Author Response · Author response to Decision Letter 2]

28 Oct 2020

Comments Page

The structure of the article must have four very clear sections: introduction, method, results and discussion and conclusion. Authors should make any necessary adjustments to provide this clarity. In this regard, authors must place point 5. justifications for using PLS within the Method section. 17

- The name of the Ethics Committee that supervised the study must be stated. This aspect is resolved. The date of the study must appear in the Method section. 17

he methodology used (The Partial Least Square-Structural Equation Model (PLS-SEM)) is not justified, nor does it seem to be specified in the text or it is not made clearly.. 17

---

## [Editor Report · Decision Letter 3]

20 Nov 2020

The Determinant Factors for the Adoption of CRM in the Palestinian SMEs: The Moderating Effect of Firm Size

PONE-D-19-33183R3

Dear Author(s),

We’re pleased to inform you that your manuscript has been judged scientifically suitable for publication and will be formally accepted for publication once it meets all outstanding technical requirements.

Kind regards,

Amira M. Idrees, Associate Professor

Academic Editor

PLOS ONE
---

## [Editor Report · Acceptance letter]

2 Dec 2020

PONE-D-19-33183R3 

The Determinant Factors for the Adoption of CRM in the Palestinian SMEs: The Moderating Effect of Firm Size 

Dear Dr. Salah:

I'm pleased to inform you that your manuscript has been deemed suitable for publication in PLOS ONE. Congratulations! Your manuscript is now with our production department. 

Kind regards, 

on behalf of

Prof. Amira M. Idrees 

Academic Editor

PLOS ONE